# Simple model for encoding natural images by retinal ganglion cells with nonlinear spatial integration

Jian K. Liu [1,2,3], Dimokratis Karamanlis [1,2,4], Tim Gollisch [1,2,5]*

1 University Medical Center Göttingen, Department of Ophthalmology, Göttingen, Germany, 2 Bernstein Center for Computational Neuroscience Göttingen, Göttingen, Germany, 3 School of Computing, University of Leeds, Leeds, United Kingdom, 4 International Max Planck Research School for Neurosciences, Göttingen, Germany, 5 Cluster of Excellence "Multiscale Bioimaging: from Molecular Machines to Networks of Excitable Cells" (MBExC), University of Göttingen, Göttingen, Germany

* tim.gollisch@med.uni-goettingen.de

**Data Availability Statement:** The spike-train data recorded for this work have been made available at https://gin.g-node.org/gollischlab/Liu_Gollisch_2021_RGC_spiketrains_spatial_contrast_model (doi: 10.12751/g-node.kod28e).

## Abstract

A central goal in sensory neuroscience is to understand the neuronal signal processing involved in the encoding of natural stimuli. A critical step towards this goal is the development of successful computational encoding models. For ganglion cells in the vertebrate retina, the development of satisfactory models for responses to natural visual scenes is an ongoing challenge. Standard models typically apply linear integration of visual stimuli over space, yet many ganglion cells are known to show nonlinear spatial integration, in particular when stimulated with contrast-reversing gratings. We here study the influence of spatial nonlinearities in the encoding of natural images by ganglion cells, using multielectrode-array recordings from isolated salamander and mouse retinas. We assess how responses to natural images depend on first- and second-order statistics of spatial patterns inside the receptive field. This leads us to a simple extension of current standard ganglion cell models. We show that taking not only the weighted average of light intensity inside the receptive field into account but also its variance over space can partly account for nonlinear integration and substantially improve response predictions of responses to novel images. For salamander ganglion cells, we find that response predictions for cell classes with large receptive fields profit most from including spatial contrast information. Finally, we demonstrate how this model framework can be used to assess the spatial scale of nonlinear integration. Our results underscore that nonlinear spatial stimulus integration translates to stimulation with natural images. Furthermore, the introduced model framework provides a simple, yet powerful extension of standard models and may serve as a benchmark for the development of more detailed models of the nonlinear structure of receptive fields.

## Author summary

For understanding how sensory systems operate in the natural environment, an important goal is to develop models that capture neuronal responses to natural stimuli. For retinal

**Funding:** TG was supported by the Deutsche Forschungsgemeinschaft (DFG, German Research Foundation) – project IDs 154113120 (SFB 889, project C01); 432680300 (SFB 1456, project B05) – by the European Union Seventh Framework Programme (FP7-ICT-2011.9.11) under grant agreement number 600954 ("VISUALISE"), and by the European Research Council (ERC) under the European Union's Horizon 2020 research and innovation programme (grant agreement number 724822). DK was supported by a Boehringer Ingelheim Fonds fellowship. The funders had no role in study design, data collection and analysis, decision to publish, or preparation of the manuscript.

**Competing interests:** The authors have declared that no competing interests exist.

ganglion cells, which connect the eye to the brain, current standard models often fail to capture responses to natural visual scenes. This shortcoming is at least partly rooted in the fact that ganglion cells may combine visual signals over space in a nonlinear fashion. We here show that a simple model, which not only considers the average light intensity inside a cell's receptive field but also the variance of light intensity over space, can partly account for these nonlinearities and thereby improve current standard models. This provides an easy-to-obtain benchmark for modeling ganglion cell responses to natural images.

## Introduction

Much of our knowledge about how neurons in sensory systems operate stems from investigations with simplified, artificial sensory stimuli, whose properties can be specifically selected depending on the research question at hand [1]. Investigating the relevance of the inferred signal processing for real-life scenarios, however, requires examining responses of sensory neurons to natural stimuli [2–5]. An important step for this transition to natural stimuli is the design of appropriate computational models for the stimulus-response relation of sensory neurons in order to capture the observed signal processing operations and test them on responses to complex or natural stimuli [6–15].

A fundamental ingredient for such models is typically the receptive field, which describes the region in stimulus space that affects a neuron's response. For retinal ganglion cells, the output neurons of the retina, spatial receptive fields are commonly used to capture how the cells respond to light spots of different sizes or to spatially structured visual stimuli, often by assuming that the cells linearly integrate signals over their receptive fields. Yet, responses under contrast-reversing spatial gratings have long revealed that many ganglion cells can display nonlinearities in their spatial signal integration [16–21].

It is thought that these spatial nonlinearities are also important under natural stimulation [22–24], even though most natural stimuli have smaller spatial contrast levels than the high-contrast reversing gratings that are typically used to study nonlinear integration and correlations in natural stimuli lead to a prevalence of low spatial frequencies and larger regions of fairly homogeneous illumination. In ON parasol cells of macaque retina, for example, spatial nonlinearities are pronounced under reversing gratings, yet nearly absent under natural stimuli [9]. Yet, sharp light intensity transitions occur also in natural visual scenes, for example, in conjunction with object boundaries, and spatial nonlinearities under natural stimuli have been demonstrated in different ganglion cells of macaque, mouse, and rabbit retina [8,9,25,26].

The source of the spatial nonlinearities appears to be the presynaptic bipolar cells [10,27–29], which provide the excitatory input to the ganglion cells and whose neurotransmitter release can show partial rectification with respect to light intensity [28,30]. This rectification occurs despite the graded, non-spike-dependent synaptic exocytosis at bipolar-cell terminals and likely follows from nonlinear dependence of synaptic exocytosis on presynaptic membrane potential and on calcium concentration [30], which may be supported by the ribbon synapse's multivesicular release [31,32]. Yet, incorporating nonlinear bipolar cell input into computational models has been difficult because determining the layout of bipolar cells and the nature of their nonlinearities either requires detailed anatomical knowledge [28] or data-intensive inference methods [10,12,33–37].

We therefore here seek a direct assessment and visualization of the importance of nonlinear spatial integration under stimulation with natural images and evaluate this for ganglion cells of

the salamander retina. Based on the observed sensitivity to spatial contrast, we then introduce a simple model that phenomenologically incorporates effects of nonlinear spatial stimulus integration and whose parameters can be obtained with relatively small amounts of data. The approach is based on identifying the receptive field of a ganglion cell and evaluating not only the (weighted) mean of light intensity across the receptive field, but also the variability of light intensity over space, measured by the (weighted) second-order statistics of the stimulus. Evaluated on recordings from salamander ganglion cells under flashed natural images, we find that this analysis reveals nonlinear spatial stimulus integration and provides a simple way to improve standard receptive-field-based models of ganglion cell activity. Furthermore, application to a dataset of mouse retinal ganglion cell recordings affirms the general applicability of the introduced analysis and modeling approach.

## Results

### Ganglion cell responses under stimulation with natural images

To investigate responses of retinal ganglion cells to flashed natural images, we projected photographic images onto isolated salamander retinas and recorded the spiking activity of ganglion cells with multielectrode arrays. The images were presented individually in a pseudo-random sequence for 200 ms each, with an inter-stimulus-interval of 800 ms (Fig 1A). Fig 1B shows an example of one of the images, overlaid with the receptive field outline of a sample ganglion cell, and the spike patterns of this cell measured for 13 individual presentations of the image. A simple measure of the response is given by the total spike count elicited by the image presentation. To obtain the spike count, we used a temporal window ranging from image onset to 100 ms after image offset. Given the response latency of around 100 ms in these neurons [35], this window typically includes all spikes elicited by the image presentation, but excludes potential spikes that might be elicited by the withdrawal of the image.

Based on this spike count response measure, we found that stimulation with different images generated diverse, yet reliable response patterns: for a given ganglion cell, the range of spike counts typically varied between zero and around 10 to 15, showing that a wide range of response strengths was elicited. For a given image, on the other hand, spike counts were highly reliable, with typical standard deviations of the spike count over repeated presentations of around one spike. Fig 1C shows the spike count responses to all 300 images for the sample ganglion cell. For each image, the variance of spike count over repeated trials was typically small and below the variability of a Poisson process (represented by the gray line), as also reflected by the per-image Fano factors (inset), which are mostly far below unity. Similar sub-Poissonian response variability had also previously been observed with artificial stimuli [38,39].

The high spike-count reliability was observed for most cells. Fig 1D shows for each analyzed cell the average spike count variance (averaged over all images) as a measure of trial-by-trial noise in comparison to the average spike count over all images and trials as a measure of the average response strength. As for the sample cell in Fig 1C, the spike count variance was on average much smaller than the average spike count for most cells, as also indicated by the average Fano factors below unity (inset), indicating sub-Poissonian noise and high response reliability for individual images. Furthermore, the noise level was generally much smaller than the signal range covered by different images, as shown in Fig 1E. For nearly all cells, the trial-by-trial variability (again measured as the single-image spike-count variance averaged over images) was much below the signal range, as measured by the variance over images of the mean spike counts.

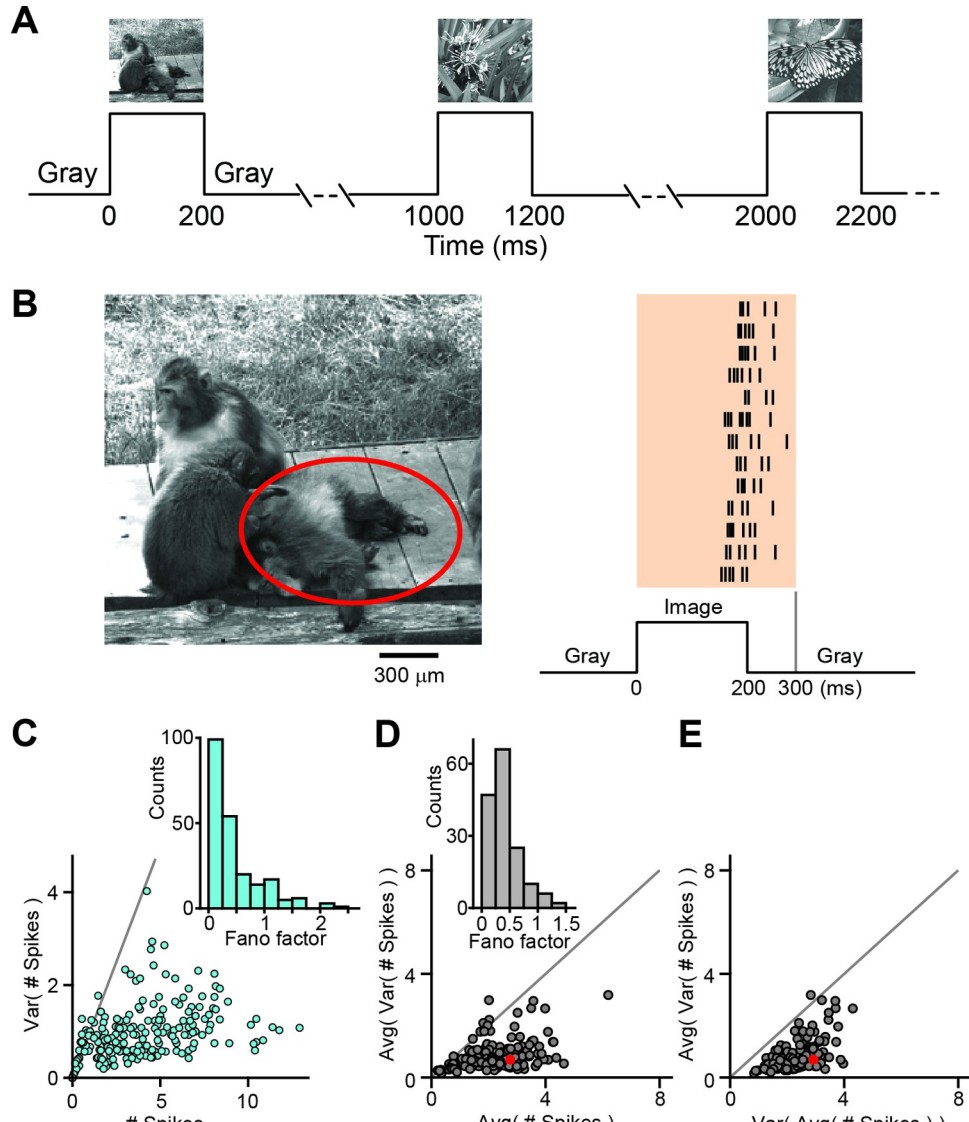

**Fig 1. Overview of salamander ganglion cell responses to natural images.** A) Schematic of the sequence of 300 natural images presented individually in a pseudo-random fashion for 200 ms each, with an inter-stimulus-interval of 800 ms. B) Left: One of the 300 natural images, overlaid with the 3-sigma outline of the receptive field of a sample retinal ganglion cell. Right: Raster plot of spike times recorded from the sample cell for 13 repeated presentations of this image. Every row corresponds to one image presentation. At the bottom, the timeline of stimulus presentation is shown, indicating the 200-ms presentation time, surrounded by periods of gray illumination. The vertical gray line at 300 ms marks the end of the applied window for counting spike numbers. C) Spike count variance (y-axis) vs. average spike count (x-axis) from the sample cell for each of the 300 images. The gray line denotes the expected relation of variance and average spike count for a Poisson process. Inset: Histogram of Fano factors for each image, excluding those with zero spikes. D) Average over images of the single-image spike count variance as a measure of response noise (y-axis) vs. the average spike count over all images as a measure of typical signal size (x-axis), shown for each recorded cell. Inset: Histogram of Fano factors for each cell, averaged over images. N = 156 cells from 9 retinas. E) Average over images of the single-image spike count variance as a measure of response noise (y-axis) vs. variance over images of the average spike count per image as a measure of response range (x-axis), shown for each recorded cell. The sample cell is marked in red in D and E.

## Local spatial contrast shapes ganglion cell responses

To assess the relation of a cell's response to the presented natural image, we tested how the cell's spike count depended on simple image statistics inside the cell's receptive field. To do so, we first determined the receptive field of a cell by standard reverse correlation [40]. The retina was visually stimulated with spatiotemporal white noise to calculate the average stimulus sequence that preceded a spike ("spike-triggered average"). Then, singular value decomposition was used to separate the spike-triggered average into a spatial and a temporal component. Finally, a two-dimensional Gaussian function was fitted to the spatial receptive field component to determine the center, size, and shape of the receptive field.

For a given image, we obtained the local stimulus for a cell by weighting the image with the 2D Gaussian representation of the receptive field. We then first considered the mean stimulus intensity ($I_{mean}$) inside the receptive field by computing the mean pixel intensity of the local stimulus, corresponding to a linear integration of the stimulus across the receptive field. Fig 2A displays the relation between $I_{mean}$ and the measured spike count for a sample cell, which can be fitted by a parameterized rectifying nonlinear function (see Materials and Methods). Together, the linear image filtering by the receptive field and the nonlinear function constitute a classical linear-nonlinear (LN) model. To assess model performance, we used the coefficient of determination $R^2$ between the spike count data and the model's prediction for the images in a test set of 150 held-out images, which were not used for fitting the nonlinear function. For the displayed sample cell, $I_{mean}$ alone already had considerable predictive power ($R^2 = 0.65$) but did not completely specify the measured spike count.

We then asked whether—beyond mean stimulus intensity in the receptive field—spatial contrast contributed to determining the spike count. To do so, we assessed for each image the local spatial contrast (LSC) inside the receptive field by computing the standard deviation of pixel values in the local stimulus. Concretely, we weighted the image with the Gaussian representation of the receptive field and then computed the standard deviation of the pixels within the 3-sigma contour of the Gaussian.

Before including the local spatial contrast in a somewhat abstract model for predicting ganglion cell responses, we aimed at directly assessing and visualizing to what extent it influences ganglion cell responses to natural images. In this respect, however, it is important to note that the LSC is not independent of the mean stimulus intensity $I_{mean}$. As depicted in Fig 2B, the LSC tended to be larger when $I_{mean}$ deviated more strongly from zero; both large positive and large negative deviations from mean light level favored a larger range of pixel intensities, as should be expected. Thus, a direct comparison of the LSC with the measured spike count, as shown for the sample cell in Fig 2C, is not suited to determine whether the LSC itself affects the cell's response. High spike counts for large LSC values and positive correlations between LSC and spike count might have simply resulted from higher $I_{mean}$ values for the corresponding images.

To test more directly for how the LSC affected the spiking response, we aimed at assessing whether it influenced the response beyond the effect of $I_{mean}$. We therefore analyzed pairs of images that, for a given cell, yielded approximately equal $I_{mean}$ values and then computed the differences in spike count and in LSC for such image pairs [26]. Concretely, we ordered all images for a given cell according to their $I_{mean}$ value and compared spike count and LSC by calculating the differences in spike counts (ΔSpikes) and local spatial contrast (ΔLSC) for each pair of neighbors in this ordered sequence. To verify that the residual differences in $I_{mean}$ did not have a major influence in this analysis, we also calculated their difference values $\Delta I_{mean}$ for the image pairs.

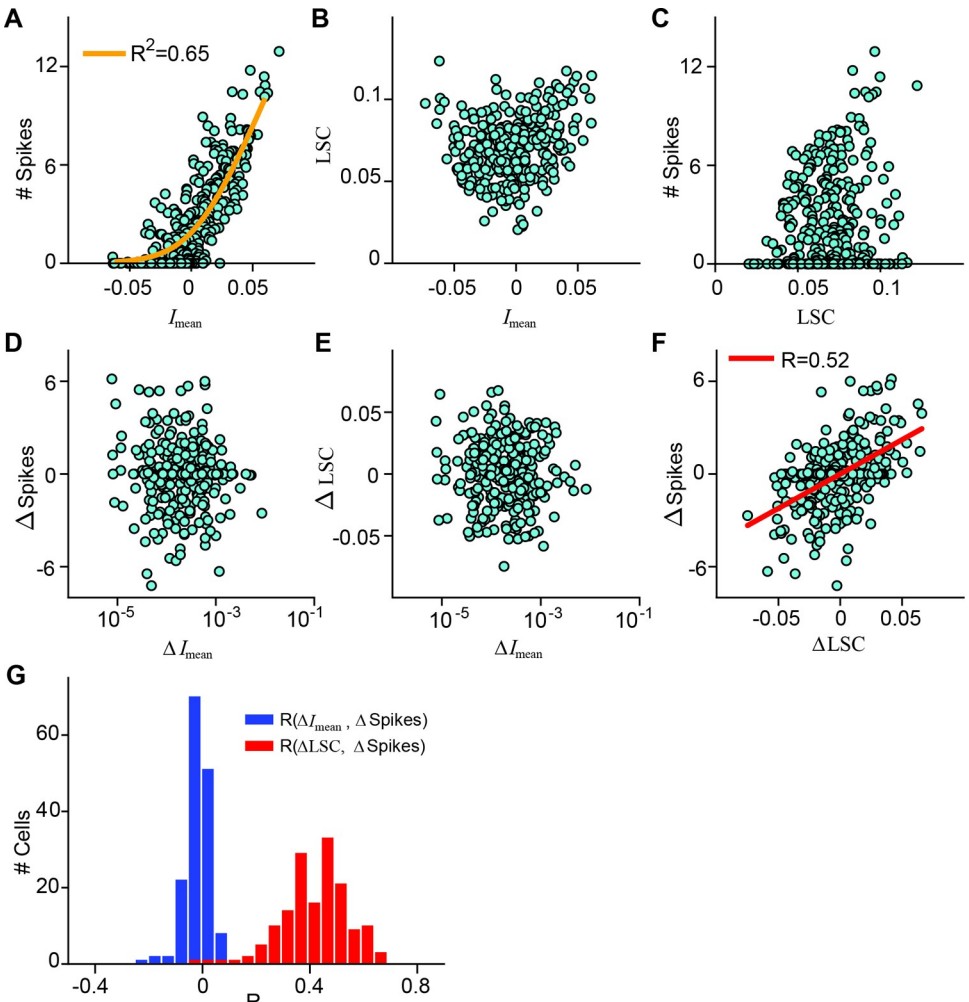

**Fig 2. Effect of local spatial contrast on ganglion cell responses.** A) Scatter plot of spike count vs. $I_{mean}$ for all 300 images for the sample cell of Fig 1B and 1C. The orange line shows the nonlinearity fitted on the 150 training images, with the stated $R^2$ value (obtained from the 150 test images) used as an evaluation of the quality of the fit. B) LSC vs. $I_{mean}$ for all images for the same sample cell. C) Spike count vs. LSC for all images for the same sample cell. D) Differences in $I_{mean}$ ($\Delta I_{mean}$) vs. differences in spike count ($\Delta$Spikes) for pairs of images that have neighboring values in $I_{mean}$, plotted for the sample cell (N = 299 image pairs). E) Same as D, but for differences in local spatial contrast ($\Delta$LSC) vs. $\Delta I_{mean}$. F) Same as D and E, but for $\Delta$Spikes vs. $\Delta$LSC. The red line is obtained by linear regression, and the Pearson correlation coefficient ($R$) is denoted in the plot. G) Distributions of the correlation coefficients $R$ for $\Delta I_{mean}$ vs. $\Delta$Spikes (blue) and for $\Delta$LSC vs. $\Delta$Spikes (red) over the 156 recorded cells.

The scatter plots of $\Delta I_{mean}$ vs. $\Delta$Spikes (Fig 2D) and $\Delta I_{mean}$ vs. $\Delta$LSC (Fig 2E) confirmed that the effects of mean stimulus intensity for the sample cell were abolished in this analysis. The residual $I_{mean}$ signal had no detectable influence on the spike count differences $\Delta$Spikes and was not correlated with $\Delta$LSC. We then found that the local spatial contrast systematically affected the spike count: $\Delta$LSC and $\Delta$Spikes displayed a pronounced correlation with $R = 0.52$ (Fig 2F). As evident from the plot, the systematic effect of the LSC for this cell was to increase or reduce responses (relative to the response determined by the mean stimulus intensity) by up to about three spikes over the range of images tested here.

To test the generalizability of the findings, we performed the same analysis on all recorded salamander OFF ganglion cells. ON cells were not considered, as these are much less frequent

than OFF cells in the salamander retina [41–43] and were only rarely encountered in our recordings. Furthermore, the few recorded ON-OFF cells were not considered here because of the additional complications arising from the (nonlinear) integration of the two parallel input pathways [44,45] and the non-monotonic contrast-response relationship. For the analyzed OFF ganglion cells, the population analysis corroborated the findings obtained from the sample cell. When pairing images with similar mean stimulus inside a cell's receptive field, the spatial contrast differences ΔLSC (but not the residual mean intensity differences $\Delta I_{mean}$) were generally positively correlated to the spike count differences ΔSpikes (Fig 2G), supporting that spatial contrast in the image can boost spike count beyond the effect of mean stimulus intensity.

## Spatial contrast model to incorporate sensitivity to spatial structure

Next, we analyzed whether this additional information contained in the LSC about the spike count can improve the response prediction over the classical LN model. To do so, we set up a spatial contrast (SC) model, which combines information from the mean intensity and from the spatial contrast inside a cell's receptive field. The processing chain is explained in Fig 3A. The spatial receptive field is obtained from responses to spatiotemporal white-noise stimulation and fitted by a 2D Gaussian. The Gaussian provides weights for each image pixel to extract the local stimulus, whose distribution of pixel contrast values yields the local mean intensity $I_{mean}$ as the mean of this distribution and the local spatial contrast LSC as the standard deviation. The model's linear activation is computed as a weighted sum of the $I_{mean}$ and LSC values (Fig 3B, left). Like in the classical LN model, the activation is turned into a prediction of the spike count through a nonlinear rectifying function. This function is fitted to the relation of the weighted sum of $I_{mean}$ and LSC and the measured number of spikes (Fig 3B, right). The weight factor that is multiplied to the LSC is an additional free parameter and is fitted together with the parameters of the nonlinearity on the training data. The model is evaluated by computing the squared correlation coefficient $R^2$ on held-out images.

Fig 3B (right) shows the relation between the activation of the SC model, as obtained from $I_{mean}$ and LSC together, and the measured spike counts for the sample cell of Fig 2. The nonlinear function captured the spike counts more accurately than in the classical LN model (cf. Fig 2A). Including the spatial contrast information improved the model performance from $R^2 = 0.65$ for the classical LN model to $R^2 = 0.76$, as again assessed on the test set of 150 held-out images.

Population analysis of the recorded OFF ganglion cells corroborated the improved model performance of the SC model. The SC model had overall considerably better performance for predicting spike counts of held-out test images than the classical LN model (Fig 3C; average $R^2 = 0.53\pm0.21$, mean±SD for LN model and $0.65\pm0.14$ for SC model; $p<10^{-6}$, Wilcoxon signed-rank test), in particular when the classical LN model originally had low performance. This is also emphasized by plotting the prediction improvement, which we calculated as the ratio between $R^2$ from SC model and $R^2$ from the classical LN model, against the performance of the classical LN model (Fig 3D, left). Furthermore, the improvement was larger when the analysis of spatial-contrast effects on spike count beyond the mean stimulus intensity (cf. Fig 2F) revealed a sizeable correlation between ΔLSC and ΔSpikes (Fig 3D, center). And the importance of the spatial contrast information for this improvement is also reflected in accompanying higher weights for the SC component in the model fits (Fig 3D, right). These results suggest that spatial contrast inside the receptive field can exert a strong effect on spike responses beyond the mean stimulus intensity, and including this information can yield strongly superior models when the classical LN model fails.

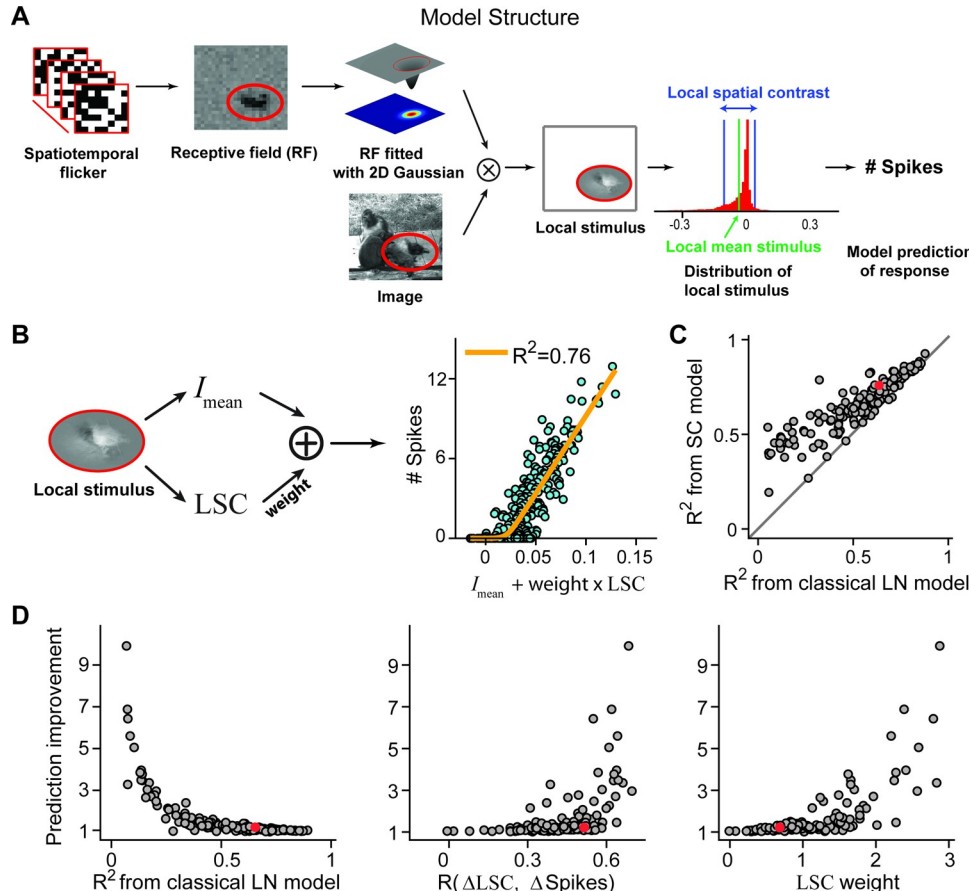

**Fig 3. Modeling responses to natural images based on simple image statistics.** A) Model for assessing the effect of simple image statistics on the spike count. Stimulation with spatiotemporal flicker is used to calculate a ganglion cell's spatial receptive field, which is then fitted by a 2D Gaussian function. Each natural image is cropped to the 3-sigma contour of the receptive field (red line) and weighted by the 2D Gaussian, yielding the local stimulus. The mean stimulus intensity ($I_{mean}$) and the local spatial contrast (LSC) are obtained as the mean and the standard deviation of the pixel intensities in this local stimulus. Both measures are used in the prediction of the spike count response. B) Left: Schematic depiction of the SC model, which uses a weighted sum of the extracted values of $I_{mean}$ and the LSC for a given image. Right: Relationship between the linear signal of the SC model and the number of spikes recorded from the sample cell of Fig 2 for all images. The orange line shows the fitted nonlinearity, with the stated $R^2$ value used as an evaluation of the quality of the fit. C) Coefficients of determination $R^2$ compared for the classical LN model, which takes only $I_{mean}$ as input, and the SC model, which takes into account both $I_{mean}$ and LSC, for all cells (N = 156). D) Relative prediction improvement of the SC model over the classical LN model (computed as the SC model performance normalized by the LN model performance) vs. model performance of the classical LN model (left), vs. the correlation coefficients calculated from ΔLSC and ΔSpikes (center), and vs. the optimal LSC weights found in the fit of the SC model. The sample cell is marked in red in C and D.

## Dependence of SC model performance on ganglion cell response type

The comparison of model performance between the classical LN model and the SC model has shown considerable diversity between individual cells. This variability raises the question whether there are systematic differences between different classes of ganglion cells. For the salamander retina, a general classification scheme of ganglion cells is still lacking [46], and physiological classifications are typically based on preferred contrast (ON versus OFF) and temporal filtering kinetics [10,41,47,48]. Following these lines, we here divided the recorded OFF ganglion cells into four groups by a cluster analysis (see Materials and Methods), according to their receptive field size and temporal filtering kinetics (Fig 4A). Two of the four classes

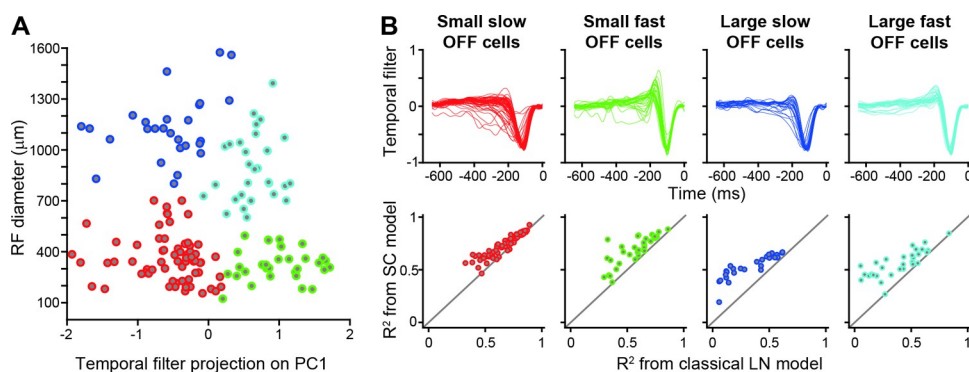

**Fig 4. Evaluation of LN and SC model separately for four functional classes of salamander ganglion cells.** A) Scatter plot of receptive-field diameter versus the projection of the temporal filter on the first principal component of all temporal filters. Temporal filters were obtained from the spike-triggered average under spatiotemporal white-noise stimulation. The colors mark the four clusters that were determined by k-means clustering. B) Collection of all temporal filters (top) and display of model performance values for the two models, separately for the four cell classes (number of cells: 66 small slow OFF cells, 33 small fast OFF cells, 27 large slow OFF cells, 30 large fast OFF cells).

had larger receptive field diameters than the other two classes, and the two classes with larger receptive fields as well as the two classes with smaller receptive field were each separated by the shape of the temporal filters. For both receptive field sizes, one class had faster kinetics with an earlier filter peak and a more biphasic filter shape.

When separating the obtained model performances according to the four cell classes (Fig 4B), we found that it was the large (fast and slow) Off cells whose response predictions benefitted the most from including spatial contrast information (prediction improvement 2.3±1.5, mean±SD, for large slow Off cells and 2.2±2.0 for large fast Off cells). The two classes of small cells, on the other hand, often showed good LN model predictions with moderate improvement from spatial contrast information (prediction improvement 1.1±0.2 for small slow OFF cells and 1.3±0.3 for small fast Off cells).

The stronger dependence of spatial contrast information in larger cells makes intuitive sense, as these cells pool information over wider spatial ranges and may therefore experience larger spatial variations in luminance. A similar relation of receptive field size and spatial contrast sensitivity is also seen in the primate retina where size as well as nonlinear effects increase from midget via parasol to upsilon ganglion cells [19].

## Application to mouse retinal ganglion cells

To test whether the SC model is also applicable to ganglion cell data from other species, we analyzed an existing dataset of responses to natural images from mouse retinal ganglion cells [26,49]. As for the salamander recordings, 300 natural images were flashed for 200 ms each, and responses were measured as the elicited spike counts. Fig 5A shows for a sample cell that the difference in spike count for pairs of natural images with similar mean luminance signals in the receptive field can be strongly correlated with the difference in spatial contrast, similar to the results for salamander retinal ganglion cells (cf. Fig 2F). For this sample cell, including information about spatial contrast in the SC model led to a slightly better fit of the spike count as compared to the classical LN model (Fig 5B).

In general, and similar to the salamander analysis (cf. Fig 3C), we observed a wide range of model performances (Fig 5C), with some cells displaying already high LN model performance and little improvement by spatial contrast information and other cells showing much better predictions with the SC model than the LN model. The distributions of model performance

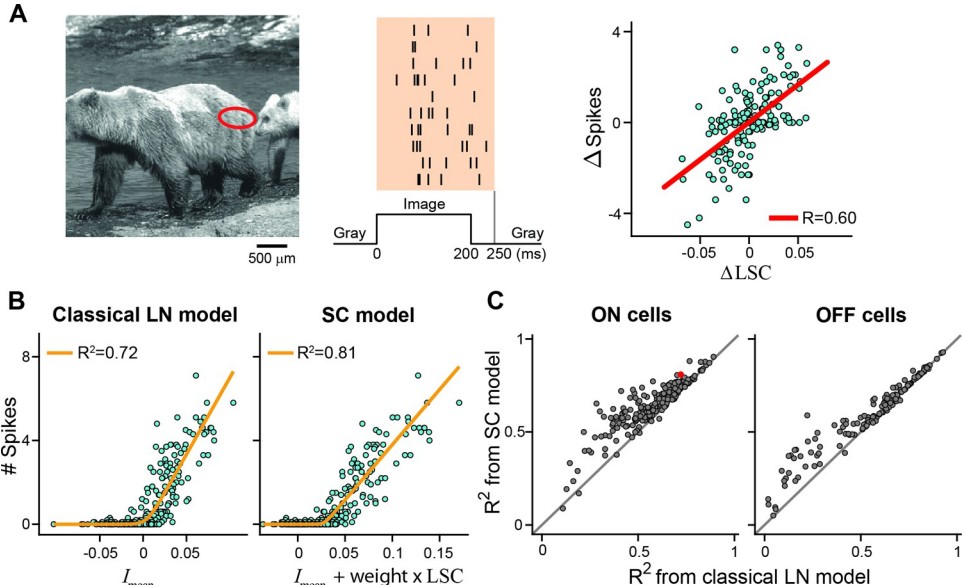

**Fig 5. Analysis of a dataset of mouse retinal ganglion cells.** A) Receptive field center (3-sigma contour, red ellipse) of a mouse ganglion cell, overlaid on a sample image (left), spiking responses of the cell to the sample image for ten trials (center), and the cell's spike-count differences versus differences in local spatial contrast for pairs of images with nearly equal mean luminance information in the receptive field (right, N = 299 image pairs). B) Scatter plots of spike count versus the linear signals of the LN model (left) and of the SC model (right) for all 300 images for the sample cell of A. The orange lines show the nonlinearities, fitted on the 150 training images, with the stated coefficients of determination ($R^2$; obtained from the 150 test images) used as evaluation of the fit quality. C) $R^2$ compared for the classical LN model and the SC model, separately for mouse ON ganglion cells (left) and OFF ganglion cells (right). The data from the sample cell is marked in red. N = 206 ON and 142 OFF cells from 9 retinas.

values indicate that predictions for some ON cells already benefitted from spatial contrast information at intermediate LN model performances around $R^2$ = 0.6, whereas predictions for OFF cells improved substantially from spatial contrast information only when LN model performance was low. Overall, however, model performance values between the two populations were similar on average ($R^2$ values for ON cells: 0.57±0.16, mean±SD, for LN model and 0.64 ±0.17 for SC model; for OFF cells: 0.55±0.23 for LN model and 0.60±0.19 for SC model). These analyses illustrate that spatial contrast information can be useful for predicting responses of mouse ganglion cells to natural images and that the SC model is directly applicable to the mouse retina.

## Assessing the relevant spatial scale of local spatial contrast

We found that the LSC can be a useful predictor for ganglion cell spiking responses. So far, we have calculated this measure based on the standard deviation of image intensities at the level of individual pixels. This takes into account intensity variations at all available spatial frequencies. Yet, it should be expected that spatial pooling by photoreceptors and bipolar cells prevents nonlinear stimulus integration for high spatial frequencies. In other words, intensity values of image pixels that are close together should be integrated linearly by the ganglion cell without any effect of how the total intensity is distributed between these pixels. Thus, there should be an optimal spatial scale for calculating the LSC, and this optimal spatial scale should be informative about the size of the relevant nonlinear subunits inside the receptive field, which are thought to correspond to bipolar cell receptive fields [10,23,27,28].

To test for the optimal spatial scale, we varied the way in which we measured the LSC and searched for the highest predictive power of the SC model. Specifically, we looked at how spatial smoothing of the image before computing the LSC affected the response prediction. We expect that image predictions improve approximately as long as smoothing occurs on spatial scales smaller than the subunits inside the receptive field, whereas smoothing at larger scales should degrade the predictions. The reason for this is that the right level of smoothing allows contrast variations on scales below the subunits to be averaged out (as we expect indeed occurs inside the subunits), whereas contrast variations that span more than a single subunit do contribute to spike count prediction and their smoothing should deteriorate model performance.

Fig 6A illustrates this spatial-scale analysis for a salamander retinal ganglion cell. The sample image on top, overlaid with the cell's receptive-field outline, was blurred by circular Gaussian filters with increasing spatial scale, ranging from 15 to 195 µm, and then pixel-wise multiplied by the cell's receptive field to yield the image patches shown below. In this way, we extracted for each image the LSC from the smoothed versions and combined this with the $I_{\text{mean}}$ from the original images to fit an SC model for each level of smoothing (Fig 6B). As before, we assessed the model quality by the $R^2$ values and evaluated the model improvement (relative to the performance of the classical LN model) as a function of the spatial scale of smoothing (Fig 6C). The spatial scale is here defined as three times the standard deviation of the Gaussian, for direct comparability with our definition of receptive field size. The optimal spatial scale is obtained as the spatial scale of smoothing for which $R^2$ is maximal, which we extracted using interpolation with a second-order polynomial fitted to the three data points around the maximum data point (green line in Fig 6C).

We performed this analysis for each recorded salamander ganglion cell. The dependence of the prediction improvement on spatial scale indeed typically showed a concave shape with a maximum somewhere between 0 and 200 µm (Fig 6D). We extracted the optimal spatial scales from the maximum for each cell and compared the findings across the four distinguished functional classes. The distributions of optimal spatial scales (Fig 6E) all show broad peaks in the range of 50–150 µm. This is consistent with the typical size of bipolar cell receptive fields in the salamander retina [50,51]. There were no obvious differences between the four functional classes (average optimal scales, mean±SD: 92±75 µm for small slow Off cells, 75±42 µm for small fast Off cells, 79±51 µm for large slow Off cells, 80±51 µm for large fast Off cells, p = 0.5 two-sided Kruskal-Wallis test). Thus, cell classes with larger receptive fields did not show larger optimal spatial scales of smoothing. For all classes, subunits may therefore be of comparable size, and larger cells have more subunits rather than larger ones. This is also emphasized when the optimal scale was normalized by the ganglion cell's receptive field size (Fig 6F). The peak positions in the distributions now differed between the four classes, and average values tended to be smaller for the classes with larger receptive fields (average relative optimal scales, mean±SD: 0.29±0.19 for small slow Off cells, 0.25±0.19 for small fast Off cells, 0.09±0.06 for large slow Off cells, 0.13±0.17 for large fast Off cells).

## Discussion

Current models of stimulus encoding by retinal ganglion cells often start with using a cell's receptive field as a spatial filter applied to incoming images. This is the case for the commonly applied LN model and for many of its extensions such as the generalized linear model [52,53] or other approaches for including spike-timing dynamics or feedback loops [54,55]. The single-spatial-filter approach remains popular because it is conceptually straightforward, amenable to simple parameter-fitting approaches [40,52,53], and remarkably successful in capturing ganglion cell responses under specific stimulus conditions or for certain subtypes of ganglion cells [6,9,26,52].

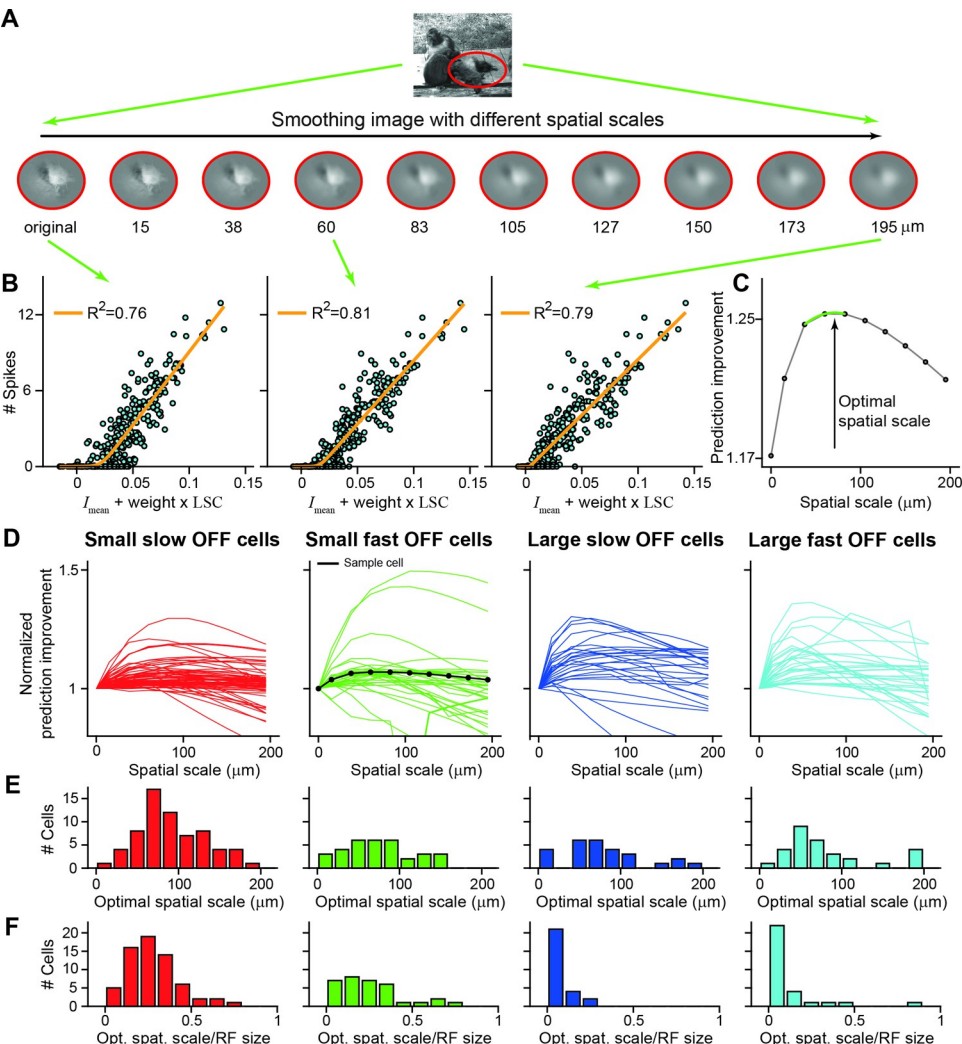

**Fig 6. Analysis of the optimal spatial scale for calculating the local spatial contrast.** A) Sample image overlaid with the 3-sigma outline of a sample cell's receptive field (red curve). Below: Local stimuli after smoothing with a 2D Gaussian filter with increasing spatial scale from 15 to 195 μm and pixel-wise weighting by the sample cell's spatial receptive field. The first image is without smoothing. B) Relation between linear signal of the SC model and measured spike count, using the same $I_{mean}$ values, but LSC values derived from the differently smoothed images, displayed here for the original image and for the images with spatial scales of smoothing of 60 and 195 μm, respectively. The orange lines show the fitted nonlinearities, and the $R^2$ values denote the corresponding model performance. C) Prediction improvement as a function of the level of smoothing for the sample cell. The optimal spatial scale is defined as the spatial scale at which $R^2$ reaches its maximum (as determined by the 2nd-order polynomial fit around the maximal data point; green line). D) Prediction improvement, normalized by the prediction improvement with no image smoothing, as a function of the level of smoothing for all cells, shown separately for the four cell classes of Fig 4. The data from the sample cell is shown in black. E) Distributions of optimal spatial scales. F) Distributions of optimal scales, normalized by each cell's receptive field size.

Yet, an underlying assumption of the simple linear spatial filtering is that all relevant non-linearities of the system can be subsumed into processing at the output stage, following after stimulus integration over space (and time) has already taken place. This clashes with the widespread finding of nonlinear spatial integration, as revealed, for example, by contrast-reversing spatial gratings [16–20]. It is thought that this nonlinear spatial integration also affects responses of retinal ganglion cells to natural stimuli, leading to failures of the model approach with a single spatial filter [7–10,12,26,56].

As the nonlinear spatial integration follows from the ganglion cell's nonlinear pooling of presynaptic bipolar cells with smaller receptive fields, the most principled way of constructing improved models has been to explicitly include this network structure by using multiple spatial filters—the model's subunits—in parallel and nonlinearly transforming the subunit signals before summation. This corresponds to a sequence of two LN models and is thus often referred to as an LNLN model. Yet, despite recent progress in inferring the layout and structure of the subunits [10,12,28,36,37,57–59], obtaining such models remains challenging, and current procedures typically require large amounts of data.

In this work, we therefore take a different approach by assessing nonlinear spatial integration through a simple phenomenological model that goes beyond the standard spatial filtering by considering the local spatial contrast inside the receptive field. We defined the local spatial contrast via the variability of stimulus intensities and found that it was positively correlated to the spike count response of the recorded cells when effects of mean stimulus intensity were removed (Fig 2). This analysis can serve as a simple and straightforward test for whether nonlinear spatial integration affects spike counts under natural stimuli, without the need to specify a concrete model of the stimulus-response relation. The measured direct effect of spatial contrast on spike count led us to a simple extension of the classical LN model by including spatial contrast as an additional model input. This yielded substantial improvements in predicting salamander ganglion cell responses to natural images, in particular for cells where the performance of the classical LN model was poor (Fig 3). Classification of the cells into different functional groups revealed that it is cells with large receptive fields whose response predictions profit most from including spatial contrast information (Fig 4), suggesting that the model improvement depends on cell type. The method's robustness to data from different species was supported by analyzing a dataset of mouse retinal ganglion cells (Fig 5), with similar findings as for the salamander retina. And finally, computing the spatial contrast measure after smoothing the image can further improve the model predictions and serve to test different spatial scales of nonlinear spatial integration (Fig 6).

The performance of the LN model and the improvement through the spatial contrast model displayed considerable variability across the analyzed population of salamander ganglion cells. The dependence on receptive field size, suggests that cell-type specificity might play a role in this variability, with larger cell types displaying stronger spatial nonlinearities, as has also been suggested for other species [19,30]. In the primate retina, for example, the smaller midget ganglion cells are more linear than the larger parasol cells and, in particular, the even larger upsilon cells [19]. More generally, differences across cell types in the characteristics of nonlinear spatial integration are a common observation. Under natural stimuli, OFF parasol cells, for example, display stronger spatial nonlinearities than ON parasol cells [9], and in the mouse retina, spatial nonlinearities in response to natural images also vary widely, with at least some of this variability depending on cell type [26].

The spatial contrast model is phenomenological and does not provide much information about the layout of subunits or the nonlinearities that act on the subunit output. The strength lies in its simplicity, requiring only measurements of the receptive field and adding just a single free parameter as compared to the classical LN model. Thus, the model may serve to assess and partly capture effects of nonlinear spatial integration when little data is available. It may also be valuable for providing a simple benchmark for comparison with more complex models of nonlinear spatial integration when the classical LN model appears too simplistic for providing a baseline measure.

Future studies may compare the performance of the spatial contrast model with full-fledged LNLN models, whose subunits of the first linear-filter stage are obtained by one of the currently developed inference techniques, including statistical analysis of spike-triggered stimuli [10,12], direct model fits [37,58], and applications from artificial neural networks [56,60]. To

date, however, obtaining a full LNLN model with inferred subunits under controlled regularization and, ideally, with optimized subunit nonlinearities is an open challenge. Besides sophisticated inference techniques, it will likely require long, dedicated recordings, potentially using spatiotemporal white-noise stimulation, to acquire suited data for parameter fitting. We expect that the spatial contrast model will be a useful tool to aid these developments by providing informative benchmarks and by illustrating the importance of spatial nonlinearities for predicting responses of a given ganglion cell.

Structurally, the proposed model is similar to a generalized quadratic model (GQM), which allows for a general quadratic function of the stimulus and a subsequent nonlinear transformation [61–63]. For the present case of natural image encoding, the stimulus is given as the set of pixel intensity values, and the general quadratic function could therefore incorporate the mean intensity via a linear component as well as the pixel standard deviation via a quadratic contribution. The difference of our approach is that we here use a particular, simple quadratic function of pixel contrast values, which does not require extensive parameter fitting, and that we use the standard deviation of pixel contrast values and thus supply the quadratic term with a square root. This provides an equal scaling of contributions from mean stimulus intensity and spatial contrast with overall stimulus contrast, which may be helpful considering the wide range of contrast values encountered in natural stimuli. A direct comparison of the spatial contrast model with a fitted GQM would be an interesting endeavor. Similar to LNLN models, however, fitting GQMs requires sufficiently large datasets and dedicated methods for regularizing or otherwise avoiding overfitting. This is of particular concern for high-dimensional stimuli, as is the case when spatial stimuli with sufficient resolution to assess contrast at fine spatial scales are considered. Selecting appropriate stimuli and constraints for the models may benefit from knowledge gained with the simpler spatial contrast model.

The simplicity of the spatial contrast model and its structural similarity to the LN model also make it amenable to different extensions, such as incorporating temporal filtering of the mean light intensity and the spatial contrast signal (either with identical or with potentially different filter shapes) and temporal feedback, such as gain control signals or post-spike filters [54,64]. Moreover, it might serve as a useful, spatially nonlinear front end in more complex cascades of cortical visual processing [65–67]. Conversely, it may help include additional, often neglected nonlinear effects in models of the retina itself. This could be used, for example, to capture nonlinear spatial integration in the receptive field surround [68], for which subunit models have been difficult to set up, or nonlinear chromatic integration [69]. Similarly, using spatial contrast directly as an additional input channel could help include nonlinear effects in the outer retina, in the transmission from photoreceptors to bipolar cells [70–74]. Such a nonlinear front end could then be combined with the typical subunit model structure that is used to capture downstream nonlinear spatial integration in the connection from bipolar to ganglion cells at the inner retina [10,12,17,18,22,23,28,37,75–77].

## Materials and methods

### Ethics statement

All experimental procedures were performed in accordance with national and institutional guidelines and were approved by the institutional animal care committee of the University Medical Center Göttingen (protocol number T11/35).

### Electrophysiology

We used retinas from adult axolotl salamanders (Ambystoma mexicanum; pigmented wild type) of either sex. Multielectrode array (MEA) recordings of ganglion cell spiking activity

were obtained as described previously [78]. In brief, after dark-adaptation of the animal and enucleation of the eyes, retinas were peeled out of the eyecup and cut in half. One retina half was placed ganglion-cell-side-down on a planar multielectrode array (Multichannel Systems, 252 channels, 10-μm electrode diameter, 60-μm spacing) and perfused with oxygenated Ringer's solution (110 mM NaCl, 2.5 mM KCl, 1.6 mM MgCl2, 1.0 mM CaCl2, 22 mM NaHCO3, 10 mM D-glucose, equilibrated with 95% O2 and 5% CO2). Recordings were performed at room temperature (20˚C-22˚C). Potential spikes were detected by threshold crossing from the amplified voltage signals after band-pass filtering (300 Hz to 5 kHz) and digitization at 10 kHz. Spike sorting was performed with a Gaussian mixture model [79]. Only well-separated units with a clear refractory period were used for further analysis.

## Visual stimulation

Visual stimuli were projected onto the retina from a gamma-corrected miniature OLED monitor (eMagin, OLED-XL series, 800 x 600 pixels with a refresh rate of 60 Hz). The monitor image was focused onto the photoreceptor layer via a telecentric lens to a pixel size of 7.5 μm x 7.5 μm. Stimuli were generated with a custom-made software, based on Visual C++ and OpenGL. All stimuli had a mean light level of 2.5 mW/m$^2$, which was also used as a background light level presented between stimuli.

Receptive fields were obtained from measurements with spatiotemporal white noise on a checkerboard layout with squares of 30 μm x 30 μm. For each square, light intensities were chosen randomly at a rate of 30 Hz from a binary distribution (100% Michelson contrast). From the recorded spikes, we computed the spike-triggered average (STA) for each recorded ganglion cell [40], taking into account stimulus sequences of 660 ms before each spike. We used singular-value decomposition [80,81] to decompose the STA into a temporal filter and a spatial receptive field and normalized each to unit Euclidean norm. Finally, we fitted a two-dimensional Gaussian function $G(\boldsymbol{x}) = A \frac{1}{2\pi\sqrt{|\Sigma|}} e^{-\frac{1}{2}(\boldsymbol{x}-\boldsymbol{\mu})^T \Sigma^{-1}(\boldsymbol{x}-\boldsymbol{\mu})} + B$ to the spatial receptive field, where $\boldsymbol{x} = (x,y)$ denotes the position in the image pixel space. The fit was obtained by least-squares optimization of the amplitude $A$, the receptive-field center position $\boldsymbol{\mu}$, the covariance matrix $\Sigma$, and the offset $B$. For further analysis (see "*Models and response predictions*" below), $G(\boldsymbol{x})$ was normalized by setting $A = 1$ and $B = 0$. The effective diameter of the receptive field was determined as $d = \sqrt{a \cdot b}$, where $a$ and $b$ are the major and minor axes of the 1.5-sigma contour of the fitted Gaussian.

To stimulate the retina with natural images, we selected a set of 300 natural photographs from the McGill Calibrated Colour Image Database [82], displaying a wide range of natural and artificial scenes and all consistent with spanning a field of view of around 20–40˚. Each image had a spatial resolution of 256 x 256 pixels, covering a total area of 1920 μm x 1920 μm on the retina. The images were converted into grayscale by a weighted average of the RGB-color channels, using a ratio of R:G:B = 30:59:11. Subsequently, all pixel values were shifted and scaled so that the mean pixel intensity of each image was equal to the background and the standard deviation was 50% of the mean intensity. Pixel values were clipped at 0% and 100% of the mean intensity to ensure compatibility with the light intensity range of the display. For each of the selected images, this occurred for fewer than 0.1% of the pixels. For all analyses, stimuli are represented by the Weber contrast $C$ at each pixel, $C = (L - L_{\text{mean}})/L_{\text{mean}}$, where $L$ is the pixel light level and $L_{\text{mean}}$ is the average light level over the image.

Images were presented individually for 200 ms each in a pseudo-random sequence, separated by 800 ms of background illumination. Responses of individual ganglion cells were quantified as the number of spikes over a 300-ms window following stimulus onset. Given the

response latency of around 100 ms, this generally excludes spikes elicited by the disappearance of the image after 200 ms.

## Models and response predictions

To assess the relevance of spatial structure in natural images for shaping ganglion cell responses, we compared two models for predicting spike counts. The first model is a classical linear-nonlinear (LN) model, which takes the cell's receptive field as a spatial filter that is applied to the stimulus. The model thus integrates light intensity signals linearly over a ganglion cell's receptive field. The second model, which we call spatial contrast (SC) model, has a similar structure as the classical LN model, but takes an additional, second input besides the linearly filtered light intensity. This second input is a measure of spatial contrast inside the receptive field, which is obtained from the standard deviation of the (weighted) pixel intensities.

Concretely, both models start with filter signals $F_{LN}$ and $F_{SC}$, respectively. For a given image, $F_{LN}$ was the mean stimulus intensity $I_{mean}$, given by the average Weber contrast as seen through the cell's receptive field. This was obtained by filtering the image with the receptive-field fit $G(\boldsymbol{x})$:

$$F_{LN} = I_{mean} = \frac{1}{N} \sum_{i=1}^{N} G(\boldsymbol{x}_i) \cdot C(\boldsymbol{x}_i)$$

where $i$ enumerates all pixel locations $\boldsymbol{x}_i$ within the 3-sigma contour of $G(\boldsymbol{x})$, $N$ is the number of these pixels, and $C(\boldsymbol{x}_i)$ is the corresponding pixel contrast.

$F_{SC}$, on the other hand, received an additional input, given by the local spatial contrast (LSC), which was computed as the standard deviation of the weighted pixel intensities:

$$LSC = \sqrt{\frac{1}{N-1} \sum_{i=1}^{N} \left( G(\boldsymbol{x}_i) \cdot C(\boldsymbol{x}_i) - I_{mean} \right)^2}$$

where $i$ and $N$ are defined as above. Note that the pixel contrast values $C(\boldsymbol{x}_i)$ are again weighted by the Gaussian profile $G(\boldsymbol{x}_i)$ of the receptive field, so that the LSC is the pixel standard deviation of the filtered image. Alternatively, the local spatial contrast could be computed as the weighted standard deviation of the original image [26], but the difference between these measures is small.

The obtained measure of local spatial contrast was added to the filtered image signal with a weight $w$ as a free parameter:

$$F_{SC} = I_{mean} + w \cdot LSC.$$

To turn $F_{LN}$ and $F_{SC}$ into predictions for natural images, we computed nonlinearities for both models from the natural images. For each model, the average responses to a training set of 150 of the natural images were used to fit a nonlinear "softplus" function of the form $r(F_X) = a_1 \cdot \ln(1 + e^{a_2 \cdot (F_X + a_3)})$, where $F_X$ stands for $F_{LN}$ or $F_{SC}$. The parameters $a_1$, $a_2$, and $a_3$ (together with the weight $w$ in the case of the SC model) were optimized according to a least-squares criterion, using the Matlab function "fminsearch". In case of the SC model, the parameters of the nonlinearity were fitted together with the weight $w$ by repeatedly alternating the least-squares optimizations of the nonlinearity and of the weight until convergence or a maximum of 500 iterations were reached. To avoid local minima, the fit was performed several hundred times with different initial values, and the solution with the minimum residual error was selected. The fitted functions were then used to obtain response predictions for the test set

of another 150 held-out natural images. To quantify model performance, we computed for each model the correlation coefficient $R$ between prediction and measured spike count and reported the explained variance $R^2$.

We recorded 9 retinas to collect 215 cells. Using the spike numbers $N_{sp}(B)$ and $N_{sp}(W)$ in response to full-field black and white stimulation ($\pm100\%$ contrast), respectively, we classified cells into 169 OFF cells with $N_{sp}(B)/N_{sp}(W)>3$, 9 ON cells with $N_{sp}(W)/N_{sp}(B)>3$, and 37 ON-OFF cells otherwise. We excluded ON and ON-OFF cells from further analyses, as these occurred much more rarely than OFF cells. We also excluded cells if the maximum of the average responses for the 300 natural images was smaller than 5 spikes, leaving us with 156 cells for the final analysis.

For classifying the cells into functional groups, we compared their receptive field sizes (measured by the effective diameter of the receptive field as explained above) and the kinetics of their temporal filters (quantified by the projection of the temporal filter obtained under spatiotemporal white-noise stimulation onto the first principal component of all temporal filters). The four groups of Fig 4A were then obtained by k-means clustering in this two-dimensional space.

### Analysis of mouse retinal ganglion cells

The analyzed data of mouse retinal ganglion cells come from a publicly available dataset [49]. Details about the applied stimuli and data acquisition can be found in the corresponding publication [26]. The natural images that had been applied to obtain this dataset were taken from the McGill Calibrated Colour Image Database [82], from the van Hateren Natural Image Dataset [83], and from the Berkeley Segmentation Dataset [84]. The images had been presented with a spatial resolution of 512 x 512 pixels in pseudo-randomized order for 200 ms each, separated by 800 ms of homogeneous illumination at background light level. Responses were measured as the spike count between image onset and 50 ms past image offset.

To classify cells into ON, OFF, and ON-OFF classes, we assessed their average spike count $R_{on}$ over all images with a net positive contrast signal $I_{mean}$ in the receptive field and the average spike count $R_{off}$ over images with negative $I_{mean}$. ON cells were defined as cells with $R_{on}>2\cdot R_{off}$, OFF cells as cells with $R_{off}>2\cdot R_{on}$, and all other cells as ON-OFF cells. For the analysis of the LN and SC models, we used information about the cells' receptive fields. These had been obtained by measuring the spike-triggered average (STA) under spatiotemporal white noise and separating the STA into a spatial and temporal component by fitting a parameterized model (see [26]) and extracting the Gaussian fit of the receptive field center from the model.

We excluded one experiment from the dataset, for which fewer than 300 images had been presented. We furthermore excluded cells for which none of the images elicited at least 6 spikes on average and for which responses to images were noisy, as detected by a symmetrized coefficient of determination of less than 0.5 between average image responses for odd versus even trials [26]. We also excluded ON-OFF cells, which would require refined models to account for the convergence of pathways and for the non-monotonic contrast-response function. These criteria yielded a dataset of 206 ON and 142 OFF cells from 9 retinas.

### Author Contributions

**Conceptualization:** Jian K. Liu, Tim Gollisch.

**Data curation:** Jian K. Liu, Dimokratis Karamanlis, Tim Gollisch.

**Formal analysis:** Jian K. Liu, Dimokratis Karamanlis.

**Funding acquisition:** Tim Gollisch.

**Investigation:** Jian K. Liu.

**Methodology:** Jian K. Liu, Tim Gollisch.

**Project administration:** Tim Gollisch.

**Resources:** Tim Gollisch.

**Software:** Jian K. Liu.

**Supervision:** Tim Gollisch.

**Visualization:** Jian K. Liu.

**Writing – original draft:** Jian K. Liu, Tim Gollisch.

**Writing – review & editing:** Jian K. Liu, Dimokratis Karamanlis, Tim Gollisch.

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
