## [Decision Letter · Decision Letter 0]

2 Nov 2021

Dear Gollisch,

Sorry for the delay -- this hit while I was on holiday. But overall positive reviews, and I think not too much work. Easy for me to say. ;)

Peter Latham, Associate editor

--------------

Thank you very much for submitting your manuscript "Simple Model for Encoding Natural Images by Retinal Ganglion Cells with Nonlinear Spatial Integration" for consideration at PLOS Computational Biology. As with all papers reviewed by the journal, your manuscript was reviewed by members of the editorial board and by several independent reviewers. The reviewers appreciated the attention to an important topic. Based on the reviews, we are likely to accept this manuscript for publication, providing that you modify the manuscript according to the review recommendations.

Sincerely,

Peter E. Latham

Associate Editor

PLOS Computational Biology

Wolfgang Einhäuser

Deputy Editor

PLOS Computational Biology

[LINK]

Reviewer's Responses to Questions

**Comments to the Authors:**

Reviewer #1: This paper by Liu and Gollisch studies the linearity of spatial integration in salamander RGCs. They build phenomenological models based on the average intensity in the receptive field of the cell and the spatial contrast for the cell responses and show that in many cases taking contrast into account improves performance. This provides an alternative to relatively complicated LNLN models to identify the subunits of RGC encoding. This is an interesting, yet somewhat marginal contribution. As the authors say in the Introduction, that we have long known that integration in RGCs can be nonlinear. The authors could bring out their contribution a bit more clearly.

The maybe most important comment is that I find this paper too focused on demonstrating the models in examples cells with the population analysis getting clearly to little attention. The results section should include a section on the number of cells, what types of cells these were (mostly OFF I presume? But fast or slow?) and so on. Also indicate the example cell in all population plots.

Minor comments

p. 2: In fact, the bipolar cell synapse can be highly nonlinear due to the ribbon nature of the synapse and multivesicular release

p. 3: a wide range of response strengths _ was_ elicited

Fig. 1C-D: Why not plot Fano Factors?

p. 5: For the LSC, where the pixels also weighted by the Gaussian profile?

p. 6: The analysis in fig. 2D-G seems fine, but overly complicated. Why not directly compare the two models with and without LSC input? That is focus on Fig. 2A and 3B.

Fig. 3C: What does “relative prediction improvement” mean?

p.7: I assume the r² values for the LSC model were also computed on a test set?

Fig. 3B: The improvement on the population level is obvious from the plot, but some statistics would be nice. What’s the average improvement? Maybe even a statistical test.

More details how the LSC weight in the LSC model was fitted should be provided.

Could the authors speculate why some neurons behave more linearly (little improvement by adding LSC) and some more non-linearly? Could that be related to different cell types (see below)? Are the cells with little performance gain by LSC tuned to high frequencies?

Suggestions for additional analysis that would add to the paper:

Does the cell type of the cell (ON/OFF, fast/slow, …) matter for the improvement in predictive performance and sensitivity for local contrast? Especially the analysis in Fig. 4F would be interesting to see by cell type.

Would the authors assume that this also works for e.g. mouse RGCs? Do they have data that could be used to show it?

It would be interesting to see how the two models discussed here compare against a standard LN model (with fitted linear filter) and a LNLN model fitted based on the data using efficient inference techniques.

Reviewer #2: This work starts from the well-known, but hard-to-model fact that retinal ganglion cells display nonlinear spatial summation of their input subunits. In current work, this can be modeled by either 1) measuring the subunit complement for each RGC directly through extensive data collection with small stimulus patches or 2) exploring a best-fit quadratic model (instead of a linear receptive field model) that also requires large amounts of data. Here, the authors propose using a particular quadratic function, the standard deviation of input intensities within the receptive field, or local spatial contrast (LSC). The manuscript shows that incorporating LSC into a standard linear-nonlinear Poisson model captures significantly more of the spiking response in salamander RGC's. By smoothing the input image, spatially and systematically, the authors show that the optimal spatial scale corresponds to about a third of the RGC receptive field size, which roughly aligns with bipolar subunit receptive field sizes in the salamander.

Overall, the work is clear, clearly presented, and clearly correct. The paper is well-written and thoroughly referenced and provides a useful tool for experimentalists to "level up" in their modeling of RGC's beyond typical, but known to be poor performing, LNP models.

The optimal spatial scale calculation in Figure 4 seems to have weak explanatory power across the cell population and one wonders whether the absolute scale quoted is statistically significant. Is any of this improved by separating by RGC sub-types in the analysis?

There is one major concern that would be worth addressing:

The LSC choice seems like a good guess at what might be a baseline relevant features for accounting for spatial nonlinearities in RGCs. However, as the authors note in the discussion, one could have used a generalized quadratic function to fit the RGC receptive field. It would be extremely useful to fit this kind of model for at least a subset of the data and show that the LSC was, indeed, the right best guess, by comparing performance between models.

Minor comment: It would be nice to include N's (156) in Figure 1 for the number of cells used in the analysis and to include a note in the caption saying how many different retinas (9?) these N cells come from, instead of just having that info in the Methods.

**Have the authors made all data and (if applicable) computational code underlying the findings in their manuscript fully available?**

Reviewer #1: **No: **The models are not available. Please make the analysis notebooks/code including fitted models available.

Reviewer #2: Yes

PLOS authors have the option to publish the peer review history of their article (what does this mean?). If published, this will include your full peer review and any attached files.

Reviewer #1: No

Reviewer #2: No

Figure Files:

Data Requirements:

Reproducibility:

References:

---

## [Editor Report · Decision Letter 1]

14 Feb 2022

Dear Gollisch,

Congratulations on a very nice paper! You nailed your replies to the reviewers, so much so that I didn't even send it back to them. If only all authors were so thorough. ;)

We are pleased to inform you that your manuscript 'Simple Model for Encoding Natural Images by Retinal Ganglion Cells with Nonlinear Spatial Integration' has been provisionally accepted for publication in PLOS Computational Biology.

Best regards,

Peter E. Latham

Associate Editor

PLOS Computational Biology

Wolfgang Einhäuser

Deputy Editor

PLOS Computational Biology

---

## [Editor Report · Acceptance letter]

3 Mar 2022

PCOMPBIOL-D-21-01608R1 

Simple Model for Encoding Natural Images by Retinal Ganglion Cells with Nonlinear Spatial Integration

Dear Dr Gollisch,

I am pleased to inform you that your manuscript has been formally accepted for publication in PLOS Computational Biology. Your manuscript is now with our production department and you will be notified of the publication date in due course.

With kind regards,

Zsofia Freund
